# Verification of the Field Productivity of *Rehmannia glutinosa* (Gaertn.) DC. Developed Through Optimized In Vitro Culture Method

**DOI:** 10.3390/plants9030317

**Published:** 2020-03-03

**Authors:** Yong-Goo Kim, Richard Komakech, Dae Hui Jeong, Yun mi Park, Tae Kyoung Lee, Ki Hyun Kim, A Yeong Lee, Byeong cheol Moon, Youngmin Kang

**Affiliations:** 1Herbal Medicine Resources Research Center, Korea Institute of Oriental Medicine (KIOM), 111 Geonjae-ro, Naju-si, Jeollanam-do 58245, Korea; kyg@kiom.re.kr (Y.-G.K.); lay7709@kiom.re.kr (A.Y.L.); bcmoon@kiom.re.kr (B.c.M.); 2Korean Medicine Life Science Major, University of Science & Technology (UST), Campus of KIOM, Daejeon 34054, Korea; richard@kiom.re.kr; 3Natural chemotherapeutics Research Institute (NCRI), Ministry of Health, P.O. Box 4864 Kampala, Uganda; 4Forest Medicinal Resources Research Center, National Institute of Forest Science, Yeongju 36040, Korea; najdhda@korea.kr; 5Department of Forest Bioresources, National Institute of Forest Science, Suwon 16631, Korea; pym5250@korea.kr; 6School of Pharmacy, Sungkyunkwan University, Suwon 16419, Korea; charmelon8@gmail.com (T.K.L.); khkim83@skku.edu (K.H.K.)

**Keywords:** *Rehmannia glutinosa* (Gaertn.) DC, medicinal plant, field productivity, drug equivalence validation

## Abstract

*Rehmannia glutinosa* (Gaertn.) DC is a perennial plant belonging to the family Scropulariidae. The root of *R. glutinosa* is used in oriental medicine and mainly grown using rootstock rather than seed cultivation, which gives rise to several problems including root rot, and results in a low productivity and poor quality. To solve the challenges involved in *R. glutinosa* seed cultivation, our team previously used the formative features and genetic analysis of *R. glutinosa* to determine the optimal in vitro tissue culture conditions for producing sterile culture seedlings and rootstocks of *R. glutinosa*. The aim of the present study was to identify differences between *R. glutinosa* standard rootstock seedlings (SR), *R. glutinosa* culture rootstock seedlings (CR), and culture seedlings (CS) under field conditions. The reproductive characteristics of the aerial part were more robust while the area and length of leaves were smaller for SR than those for CR and CS. The characteristic that differed the most in SR was flowering, which did not occur in CR and CS. In addition, the fresh and dry weights of the subterranean parts of CR and CS were two-fold greater than those of SR. Fourier transform near-infrared (FT-NIR) analysis showed only slight differences between the chemical constituents of SR and its culture products, which was confirmed by measuring the content of catalpol, an indexing substance. Catalpol had a reduced content in the culture products compared to SR. However, this difference was not significant. Our findings will be useful for the identification of the best seedling type of *R. glutinosa* to enable its mass production.

## 1. Introduction

*Rehmannia glutinosa* is a perennial herb belonging to the Scrophulariaceae family. *Rehmannia glutinosa* f. *huechingensis* is a plant that originated from *R. glutinosa* and is cultivated in China. *Rehmannia glutinosa* and *R. glutinosa* f. *huechingensis* are not taxonomically classified as the same species [1]. The roots of *R. glutinosa* are thick and yellowish in color, while the stems have grayish short hairs. The leaves appear to be gathered at the roots with edges that are blunt and serrated. The flowering form is raceme and blooms purple flowers from June to July [2]. The root of *R. glutinosa* is an important resource used in herbal medicine. In oriental medicine, *R. glutinosa* is processed and used in different forms including fresh raw and dried forms [1,3] to treat hemostasis, fever, diuretic, and hypoglycemia in humans [4]. Recently, *R. glutinosa* has been reported to have anti-obesity properties [5,6]. *Rehmannia glutinosa* is one of the main ingredients of Kyung-Ok-Ko, which is an oriental medicine [7].

The main components of *R. glutinosa* are β-sitosterol and mannitol even though it also contains traces of stigmasterol and campesterol along with other secondary metabolites such as rehmannin, catalpol, alkaloids, and fatty acids [8,9]. A number of phytochemical compounds such as iridoids, ionone glycosides, iridoid glycosides (e.g., Catalpol), saccharides, sesquiterpenes, phenylethanoid glycosides, cerebrosides, norcarotenoids, and phenethyl alcohol glycosides have been isolated and identified from various extracts including methanol, hexane, ethanol, acetone, and ethyl acetate extracts and from different forms of roots (fresh, steamed, and dried) of *R. glutinosa* [10,11,12]. The compounds were isolated by different chromatographic techniques and their chemical structures determined through extensive spectroscopic techniques including UV, IR, HR-ESI-MS, 1D, and 2D NMR, use of CD data and chemical methods [11,12,13]. The most abundant compounds in *R. glutinosa* are iridoid compounds with more than 33 iridoid monomers including acetylcatalpol, dihydrocatalpol, danmelittoside, rehmaglutin A, B, C, D, rehmannioside A, B, C, D, and rehmaglutin isolated and identified [10,11,12]. Important saccharides including five forms of oligosaccharides such as verbascose, stachyose, mannitol and sucrose, mannotriose, and raffinose have been isolated [10]. Catalpol, which is the main active chemical compound in *R. glutinosa*, is an iridoid glycoside that possesses diverse therapeutic properties such as anti-inflammatory effects and reduces bleeding tendency, protects the liver against damage, boosts the production of sex hormones, and reduces blood sugar [8,10,11,12]. Catalpol is selected as an *R. glutinosa* quality-control component and was incorporated in Chinese Pharmacopoeia in 2015 [8].

*Rehmannia glutinosa* is an allogamous plant, such that cultivation methods using rootstocks, which form a type of nutrient breeding methods, are commonly used for the production of plant materials of uniform quality [14]. Rootstocks, which are vertical roots, can be of various lengths and thicknesses. However, rootstocks with a length of 4–8 cm and a thickness of 0.5–2 cm are commonly used for propagation [15,16]. The cultivation of *R. glutinosa* rootstocks is associated with a number of problems, which result in an unstable production, including exposure to diseases or viruses, such as root rot disease, during plant growth or storage.

To resolve these issues, studies on *R. glutinosa* tissue culturing have been carried out in relation to callus-induction and subsequent regeneration and on seed germination aspects [14,15,16,17,18,19,20]. In addition, somatic embryogenesis via suspension and cultivation using *R. glutinosa* callus and the production of artificial rootstock and seedlings have been reported. However, the majority of studies on the previously mentioned techniques involve small-scale culture systems and, thus, might not translate to a large-scale in vitro culture. Our research team was able to establish the optimal culture conditions for *R. glutinosa* culture seedling and culture rootstock production for the in vitro culture of *R. glutinosa* in our previous work [21].

In a previous study, our team determined the morphological characteristics and origins of several varieties of *R. glutinosa* using genetic analysis to develop an optimized culture system using various medium conditions and treatments with various plant growth hormones [21]. The present study is a follow-up study to investigate the hypertrophy and enhancement of the biomass of medicinal parts of *R. glutinosa* cultivated in the field using the mass proliferation system developed in our previous work.

To enable the commercialization of the method of *R. glutinosa* cultivation developed in our previous study, its application in the field and comparative studies of products derived from *R. glutinosa* cultivated using this method with existing products are needed. The purpose of this study was to compare the growth characteristics of *R. glutinosa* plants in the field cultivated using the method developed by Kang et al. [21] and *R. glutinosa* existing currently in the market to compare the productivity of the medicinal parts of the plants. Additional experiments were conducted to analyze the medicinal components of *R. glutinosa*. The results presented in this study will provide a verifiable strategy for the mass production of the medicinal plant *R. glutinosa* in the field.

## 2. Results

### 2.1. Comparison of the Plant Growth Characteristics of Standard Rootstock Seedling (SR), Culture Rootstock Seedling (CR), and Culture Seedling (CS)

#### 2.1.1. Aerial Part

By comparing the growth characteristics of the aerial parts of the SR, CR, and CS, *R. glutinosa* seedlings cultivated over a period of three months, only SR seedlings flowered (Figure 1). First, we observed that the number of leaves for SR (47.4 on average) was significantly greater than those for CS and CR (both had 28.5 leaves on average) (Figure 2). The width of the leaves was the highest for the CS seedlings, which averages 12.9 cm, while the width of leaves for the CR seedlings was 10.5 cm on average. The SR seedlings had the lowest length of 6.7 cm on average (Figure 2). The leaf widths were significantly different among all seedlings. The lengths of the leaves were similar to the widths of the leaves. CS had the highest leaf width (25.8 cm on average), which was followed by CR (21.8 cm on average) and SR (15.6 cm on average). The lengths of the leaves were not significantly different between CS and CR, but the lengths of the leaves of both CS and CR showed significant differences compared to that of SR.

The rate of plant death was highest (17.6%) for CS, which was followed by CR (11.1%) and SR (8.1%) (Table 1). The number of inflorescences was found to average 1.9 peduncle for SR, with 7.7 cm being the average length. Inflorescences and peduncles were not identified for CR or CS (Table 1).

#### 2.1.2. Subterranean Part

The growth characteristics of the subterranean parts of the *R. glutinosa* seedlings were investigated. The subterranean part of the *R. glutinosa* seedlings is used in oriental medicine. The seedlings were harvested in December, which is the harvest time of *R. glutinosa*. The fresh and dry weight of the roots were then examined.

The average fresh weight of the subterranean part was the highest for CR (294.8 g), which was followed by that for CS (256.1 g) (Table 2). SR had the lowest weight at 118 g. The fresh weight of the entire subterranean part was higher for CR and CS than for SR. The total dry weight of the subterranean part was also the highest for the CR seedling (Figure 3).

The length of the *R. glutinosa* tuberous root was found to be the highest for SR, while CS was found to have the shortest root length. Statistical differences were only significant between SR and CS root lengths (Figure 4). The fresh weight of the tuberous root was highest for CR seedlings, while SR seedlings had the lowest weight. The dry weight of the tuberous root was found to be proportional to the fresh weight. The surface area of the tuberous root was the highest for CR (46 cm^2^). The second highest area was measured for CS at 39.6 cm^2^. SR was found to have the lowest surface area among the three seedling types with an average surface area of 30.3 cm^2^ (Table 2).

### 2.2. Multivariate Statistical Analysis and Component Analysis Using FT-NIR and LC/MS (Liquid Chromatography/Mass Spectrometry)

To use the roots of two types of *R. glutinosa* plants (CR and CS) developed through tissue culture under optimized conditions (hormones, etc.) as raw materials for the preparation of drugs, the identification of the raw materials was required to determine the differences between existing CS in circulation and CS developed using the cultivation method developed by us. Using Fourier Transform Near-Infrared Spectrometer (FT-NIR), the result of the Bioequivalence Test was verified. The FT-NIR analysis results showed the same pattern for the peaks of three types of seedlings (SR, CR, and CS) (Figure 5C).

The three *R. glutinosa* seedling types were investigated for drug equivalence validation by identifying their chemical profiles using LC/MS analysis. In particular, the main bioactive component, catalpol, was assessed (Figure 5A,B). Each sample was extracted using 70% MeOH with reflux to obtain a crude MeOH extract, which was analyzed by LC/MS analysis to determine the chemical profile. The LC/MS analysis of the extracts demonstrated that the chemical profiles of the four tested extracts were very similar (Figure 5D). Catalpol was detected in a UV chromatogram obtained through LC/MS performed at 254 nm and 315 nm. The similarity between the chemical profiles determined through LC/MS analysis indicated the drug equivalence of the three types of *R. glutinosa* seedlings.

To determine the catalpol concentrations reliably for the three seedling types, a calibration curve was plotted for catalpol. A range of five concentrations was used in triplicate to obtain a regression equation and to assess the response linearity (*R*^2^), limit of detection (LOD), and limit of quantification (LOQ). Linearity was evaluated by determining the *R*^2^ value from the equation obtained by the measurement of peak areas at five serially diluted concentrations. The LOD and LOQ were also sufficiently low (Table 3). By analyzing the MeOH extracts of the three seedling types of *R. glutinosa,* the levels of catalpol in the seedlings were determined to have the following order: SR (16.88 µg/mL) > CR (13.21 µg/mL) > CS (8.83 µg/mL). The quantity of catalpol in SR was relatively higher than that in CR.

## 3. Discussion

Our research team has previously published a research paper on the establishment of optimized organizational culture conditions for *R. glutinosa* seedlings [21]. Using the CS and CR seedlings developed under optimized hormone conditions determined in this previous study, in the current paper, we carried out comparative experiments on biomass amplification of the medicinal parts and plant growth characteristics of SR currently being grown in the field. The technology for mass production through tissue incubation of *R. glutinosa* involves regeneration plants using the adventitious root. The cultivation of *R. glutinosa* seedlings using a bio-reactor has already been carried out in previous studies [22,23].

Comparing the growth characteristics of the aerial parts of three *R. glutinosa* seedling types, SR, CR, and CS, a greater number of leaves was observed for SR compared to those for CS and CR. However, we were able to clearly observe with the naked eye that the surface areas of the leaves of CS and CR were significantly larger than those of the leaves of SR. The area and number of the leaves of a plant are closely related to its photosynthesis efficiency. An increased photosynthesis efficiency has a significant effect on plant root development [24,25]. Photosynthesis is a very important process, especially for medicinal crops like *R. glutinosa*, since the subterranean part of these plants, and not the aerial part, is used as the medicinal material. Plant growth hormones play an important role in the development of plants. Auxin and cytokinin, which are two of the major plant growth hormones (PGR), are known to be involved in cell division, expansion, and differentiation. For CR and CS, plants obtained through an in vitro culture of seedlings at the optimal hormone concentrations [26,27,28] established by our research team, the areas of the leaves were larger than those of leaves produced by plants obtained through an in vitro culture of SR currently used in the field (Figure 2). Auxin is known to induce plant vascular differentiation [29]. It is believed that leaves with increased levels of auxin develop more fascicular tissue, which increases physiological activity, such as active photosynthesis, which affects the development of the leaf area. As a result, it is believed that the development of subterranean roots may also have been affected by an increase in auxin in the seedling culture medium. According to early studies on the flowering of plants in the presence of light, the factor that induces flowering moves from the leaves to the top of the plant. This factor is a hormone known as florigen. However, despite decades of research, it is yet to be fully characterized. It is a common belief that flowering is caused by the interactions among various hormones. The most prominent morphological feature of SR, CR, and CS was that, for the *R. glutinosa* seedlings of the CR and CS types cultivated at optimized hormone concentrations, we found that the corresponding field plants did not flower (Figure 1 and Table 1). This phenomenon is closely related to auxin, which is a plant hormone. However, auxin is known to not be involved in the promotion of flowering. Rather, treatment with auxin inhibits the formation of flowers [30,31]. However, the suppression of blooms is thought to be a secondary effect, involving the induction of ethylene production by auxin in some cases. For example, in pineapples, it has been reported that flowering time can be adjusted by treating plants with auxin and ethylene [32]. Therefore, when CR and CS are grown in the field, auxin suppresses blooming. Flowering anthesis is a very important part of harvesting *R. glutinosa* for medicinal use. The stalks of flowers that form during the blooming season make the harvesting of *R. glutinosa* more difficult since they need to be removed after harvesting. Therefore, the CR and CS developed by our research team provide the advantage of suppressed flowering, which will reduce the amount of labor and expenses involved in the harvesting of the medicinal materials of this plant.

The production of the subterranean part of *R. glutinosa* is very important since the root represents the medicinal part of this plant. Studies using plant hormones to increase biomass production or enlarge plant roots have also been performed for *Polygoum multiflora*, which is a herbal plant whose roots are also used in traditional medicine [33]. In addition, studies have been carried out on the thickening of roots by hormone processing and biomass enhancement based on sugar content [34]. Our results regarding the reproduction of *R. glutinosa* roots for three types of seedlings (SR, CR, and CS) are as follows. The fresh weight and dry weight of the whole subterranean part of *R. glutinosa* showed a two-fold increase in biomass for CR and CS compared to those for SR (Table 2). When comparing the root tubers, which is the part of the plant used as a medicinal ingredient between the seedlings, we found that the length of tubers did not differ significantly between the three seedling types. However, the fresh weight and dry weight of the root tubers for CR and CS were greater than those of the root tubers of SR. For CR, we found that the fresh and dry weights were over two-fold higher than those for SR (Figure 4). These differences result from treatment with plant hormones. The adventitious root originates from mature cells that have re-acquired cell division activity within various tissues [33,34]. These meristematic cells develop into calyptrogen in a way that is similar to the development of cells in the lateral root. The stimulating effect of auxin, which forms adventitious roots in plants, is known to be very useful in nourishing plants. In the case of CR and CS developed under optimized in vitro culture conditions, the biomass increased due to a greater active reproduction than that of SR, which results from the effect of auxin.

The FT-NIR results showed that the *R. glutinosa* developed using the three types of seedlings had similar structures used as raw materials, which indicates that the roots of the *R. glutinosa* developed by our research team are viable for medicinal use. For the LC/MS results, the SR seedlings had the highest catalpol content due to the fact that, in most plants, secondary metabolites are highly secreted under biotic or abiotic stress conditions [35,36]. As such, the levels of catalpol in the two types of *R. glutinosa* seedlings (CR and CS) grown under optimal culture conditions were low. However, the difference in catalpol content was negligible. Since the bioactive components of *R. glutinosa* comprise not only catalpol but also verbascoside, our results will need to be verified in future studies [1,3,37]. Furthermore, high levels of catalpol could induce side effects when using the root of *R. glutinosa* for medicinal purposes. Thus, this needs to be taken in consideration when cultivating *R. glutinosa* for use in oriental medicine.

Although the results of the FT-NIR and LC/MS analyses showed a close similarity between the commercialized *R. glutinosa* standard rootstock seedlings (SR) and the culture rootstock seedlings (CR), our results will need to be interpreted in combination with those of previous research and possible hypotheses. Their implications and these findings are discussed in the widest context possible, such that future study directions may also be strong.

In conclusion, we found, in this case, that, when *R. glutinosa* CR and *R. glutinosa* CS, were grown in the field under in vitro culture conditions (phytohormone control, etc.), the mass production of the medicinal parts of *R. glutinosa* was optimized. The mass production of the subterranean parts of the CR and CS seedlings, which are used for medicinal purposes, was twice as high as that of the subterranean part of SR in circulation. The content of the indicator for the verification of bioequivalence, catalpol, was found to be lower in CR than in SR. However, the difference was not significant. In addition, the production ratios of the subterranean part were higher CR and CS than that for SR. As such, CR and CS provide an advantage in terms of reducing the amount of labor needed for obtaining *R. glutinosa* medicinal parts since they did not undergo stalk development. Further research involving additional field tests in different geographical regions will need to be performed. However, the results presented in this scenario provide strategical insights for the field of efficient mass-production research for the medicinal crop *R. glutinosa*.

## 4. Materials and Methods

### 4.1. Raw Materials: Plant Materials and Transplanting Conditions

The aim of this experiment was to validate the productivity of *R. glutinosa* (a herbal medicine resource plant) in the field. The experiments were conducted at the National Forest Medicinal Resources Research Institute located in Punggi-eup, Yeongju, Gyeongsangbuk-do, South Korea. Three types of seedlings, which include SR (standard rootstock seedling), CR (culture rootstock seedling), and CS (culture seedling), were grown under conditions suitable for their transplantation into the field and then into soil for 40 weeks each. The transplanted soil area was divided into three 6 m × 6 m areas. The three types of seedlings were randomly transplanted with intervals of 15 cm × 15 cm between them. The total number of transplanted plants was 40 for each type. On the first day after transplantation, the seedlings were watered to allow them to adapt to the soil. Afterward, no watering or fertilizer and pesticide treatments were given.

The intervals between plants were 15 cm, from the left to the right. The plants were harvested in November of the same year. After harvesting, the levels of production and growth conditions were determined for the SR, CR, and CS seedlings (Figure 6).

### 4.2. Characterization of R. Glutinosa Plant Growth

The growth characteristics of the aerial parts of the SR, CR, and CS *R. glutinosa* seedlings were investigated in July. A survey of the plant survival rates and the characteristics of flowers was conducted in September (Figure 1). To determine the growth characteristics of the three types of *R. glutinosa* seedlings planted, each seedling type was divided into two parts: aerial part and subterranean part. The study of the aerial part was conducted in July two months after transplantation. In the case of the subterranean part, the study was conducted in November during the harvest season.

#### 4.2.1. Assessment of the Aerial Part of *R. glutinosa*

For the aerial part, we examined the number of leaves, the width and length of the leaves, the number of base shoots, and the number of rhizome shoots. To determine the survival rate, the number of plants that were harvested and the number of plants that were killed were investigated. The numbers of inflorescences and lengths of peduncles were also determined.

#### 4.2.2. Assessment of the Subterranean Part *R. glutinosa*

In the subterranean part of the *R. glutinosa* seedlings, the length of the tuber and the fresh weight of tuberous root after harvesting were measured using a scale. The dry weights of the tuberous roots were measured after drying the harvested roots completely for three days using a dry oven at 150 °C to ensure there was no moisture in the roots. To measure the exact surface area of the root used for medicinal purposes, the root area was measured by dividing it into three parts known as SA, PA, and Vol, using WinRHIZO. The roots were spread out on a root-positioning tray and scanned using a flatbed scanner (GT-20000, EPSON, Nagano, Japan). The images obtained were analyzed using WinRHIZO Pro software, v.2005a (Regent Instruments, Quebec, QC, Canada). Lastly, the overall fresh and dry weights of the aerial and subterranean parts were measured and compared.

### 4.3. FT-NIR and Quality Inspection

FT-NIR spectra were recorded on a TANGO FT-NIR spectrometer (BRUKER Corporation, USA) equipped with an NIR (Near-infrared) fiber-optic probe (Type 847-072200), an interferometer, an InGaAs detector, a wide band light source (50 W), and a quartz halogen to provide interactance measurements. The *R. glutinosa* samples were placed on a holder, which ensures the stem-calyx axis was horizontal. For each sample, the interactance spectrum was measured for three opposite, equatorial positions, and the averaged spectrum per sample was used for analysis. At the head of the bifurcated cable, the source and detector fibers were placed randomly. Light was guided to the sample using source fibers and, from the sample, using detector fibers to a Tango FT-NIR spectrometer with a spectral range of 800–2500 nm. The mirror velocity was 0.9494 cm s^−1^ and the resolution was 16 cm^−1^ in this experiment. To avoid surface reflectance and guarantee subsurface penetration of the light into the *R. glutinosa* samples, the bifurcated optical probe was placed at a 75° angle to the level.

### 4.4. Extraction

The extracts obtained from the *R. glutinosa* standard rootstock (SR) (5.0 g), *R. glutinosa* culture seedlings (CS) (5.0 g), and *R. glutinosa* culture rootstock seedlings (CR) (5.0 g) were used for the LC/MS analysis. Each of the samples was ground and extracted using 70% MeOH, refluxed for 90 min, and filtered. The resultant solution was then filtered using Whatman filter no. 1 and concentrated under a vacuum with reduced pressure at 40 °C and 55 rpm using EYELA N-1200B (Tokyo Rikakikai Co. Ltd., Japan) in an efficient rotary evaporator to obtain the MeOH extract. This extract was subsequently used for LC/MS analysis.

### 4.5. LC/MS Analysis for Drug Equivalence Validation

LC/MS analysis was performed on an Agilent 1200 Series analytical system equipped with a photodiode array detector integrated with a 6130 Series electrospray ionization (ESI) mass spectrometer. Each crude extract of the samples (1.0 mg) was first dissolved in 50% aqueous methanol solution (1.0 mL) before adding additional methanol to make the total volume up to 100 μg/mL. The solution was then filtered using a 0.45-mm hydrophobic polytetrafluoroethylene (PTFE) filter. The resultant filtrate was analyzed using an LC/MS with a Kinetex C18 column (2.1 × 100 mm, 5 μm, Phenomenex, Torrance, CA, USA) at 25 °C. Formic acid in water [0.1% (*v*/*v*)] (A) and methanol (B) were used as the mobile phase at a flow rate of 0.3 mL/min based on a programmed gradient elution of 10–90% (B) for a duration of 30 min, which was followed by 100% (B) for 1 min, 100% (B) isocratic for 10 min, and 10% (B) isocratic for 10 min for post-run column reconditioning.

### 4.6. Quantitative Analysis of Catalpol

The detection of catalpol was analyzed using an LC/MS, Agilent 1200 Series analytical system equipped with a photodiode array detector, and a 6130 Series ESI mass spectrometer. The calibration curve and linear regression equation were generated using an external standard of catalpol. A standard stock solution of catalpol was prepared in MeOH at 1.0 mg/mL. The working solutions were composed of a mixture of the stock solution after serial dilutions with methanol, which results in five solutions with varying concentrations. The working solutions were then filtered through a 0.45-mm hydrophobic polytetrafluoroethylene filter prior to LC/MS injection. Linearity was plotted through linear regression analysis using the integrated peak areas (*Y*) against the concentration of each standard (*X*, µg/mL) at five serial dilution concentrations. The crude MeOH extract (1.0 mg) of the *R. glutinosa* seedlings was dissolved in MeOH (1.0 mL), which generates the sample stock solution, and was further diluted with MeOH to provide a solution of 100 µg/mL. The solution was filtered through a 0.45-mm hydrophobic polytetrafluoroethylene filter and, lastly, LC/MS analyzes the solution using a Kinetex C18 column (2.1 × 100 mm, 5 µm, Phenomenex, Torrance, CA, USA) at 25 °C. The mobile phase consisted of formic acid in water [0.1% (*v*/*v*)] (A) and methanol (B), which was delivered at a flow rate of 0.3 mL/min using the following programmed gradient elution: 10–90% (B) for 30 min, 100% (B) for 1 min, 100% (B) isocratic for 10 min, and 10% (B) isocratic for 10 min for the post-run reconditioning of the column. The injection volume was 10 µL. The quantification of catalpol was based on the peak area obtained through mass spectrometry (MS) detection in the selected ion monitoring (SIM) mode and catalpol levels were calculated as equivalents of the standard. Content was expressed as micrograms per 1 mg of extract weight. Sensitivity was assessed by determining the LOD and LOQ. LOD and LOQ were determined using the signal-to-noise (S/N) method, where 3 and 10 were used as S/N ratios for LOD and LOQ.

### 4.7. Data Recording and Statistical Analysis

Data were subjected to analysis of variance and mean separation at a significance level of *p* = 0.05 with ANOVA and Bonferroni’s post-hoc test using GraphPad Prism (GraphPad software, Ver. 5. 03) and Statistical Package for the Social Sciences (SPSS) statistics (IBM Corporation, V. 20.0 Armonk, NY, USA).

## 5. Conclusions

In conclusion, we found here, that when *R. glutinosa* CR and *R. glutinosa* CS were grown in the field under in vitro culture conditions (phytohormone control, etc.), the mass production of the medicinal parts of *R. glutinosa* was optimized. The mass production of the subterranean parts of the CR and CS seedlings, which are used for medicinal purposes, was twice as high as that of the subterranean part of SR in circulation. The content of the indicator for the verification of bioequivalence, catalpol, was found to be lower in CR than in SR. However, the difference was not significant. In addition, the production ratios of the subterranean part were higher CR and CS than that for SR. As such, CR and CS provide an advantage in terms of reducing the amount of labor needed for obtaining *R. glutinosa* medicinal parts since they did not undergo stalk development. Further research involving additional field tests in different geographical regions will need to be performed. However, the results presented in this case provide strategical insights for the field of efficient mass-production research for the medicinal crop *R. glutinosa*.

## Figures and Tables

**Figure 1 plants-09-00317-f001:**
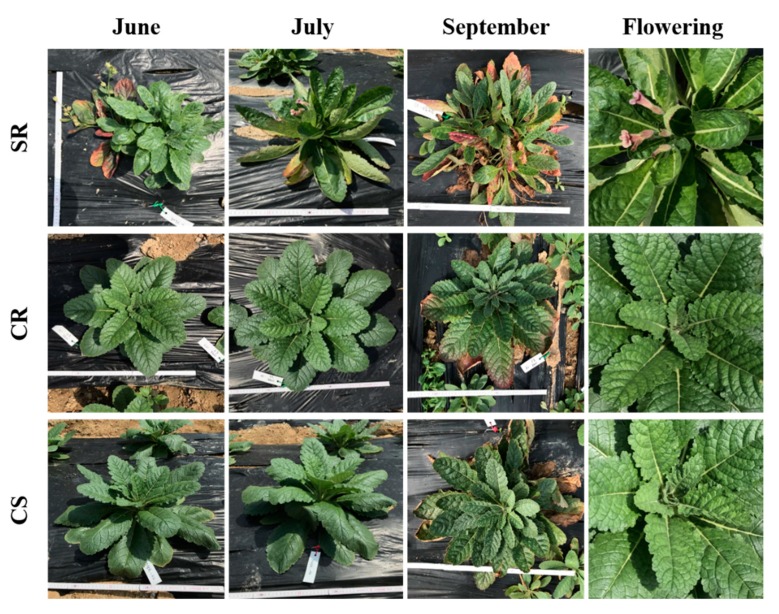
Comparison of the development and flowering of the aerial parts of the three types of *R. glutinosa* seedlings. SR, standard rootstock seedling. CR, culture rootstock seedling. CS, culture seedling.

**Figure 2 plants-09-00317-f002:**
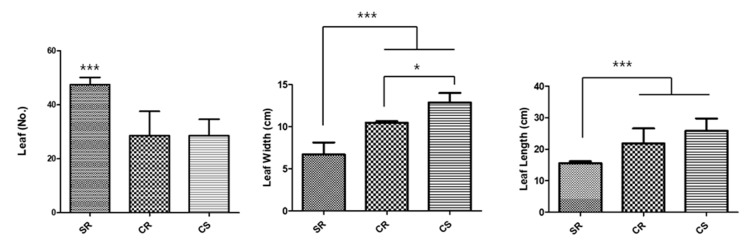
Comparison of the development of the aerial parts (number of leaves, leaf width, and leaf length) of three types of *R. glutinosa* seedlings (SR, CR, and CS). The indicated statistical differences were calculated using two-way ANOVA and Bonferroni’s post-hoc test: * *p* < 0.05, *** *p* < 0.001. N = 5.

**Figure 3 plants-09-00317-f003:**
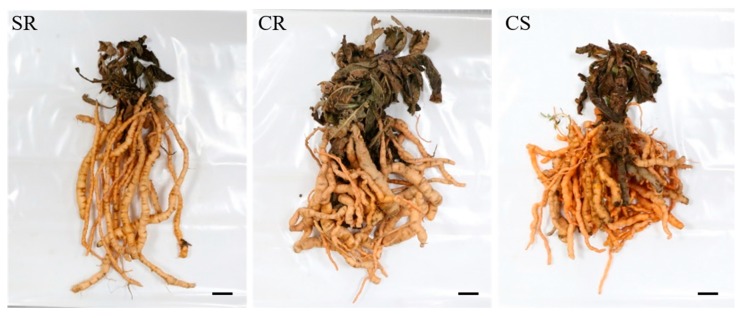
Comparison of root morphology of the different seedling types. SR, Standard Rootstock seedling. CR, Culture Rootstock seedling. CS, Culture seedling. The scale bar indicates 1.0 cm.

**Figure 4 plants-09-00317-f004:**
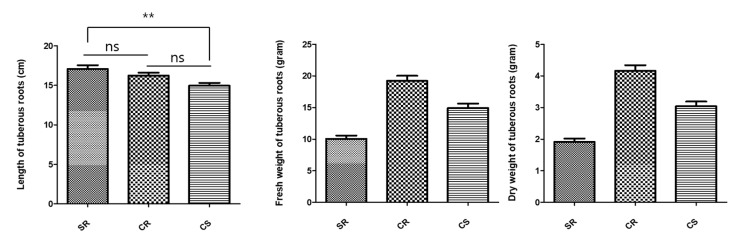
Comparison of the development of the subterranean parts of the three different types of *R. glutinosa* seedlings (SR, CS, and CR). The indicated statistical differences were calculated using two-way ANOVA and Bonferroni’s post-hoc test: ** *p* < 0.01. N = 5.

**Figure 5 plants-09-00317-f005:**
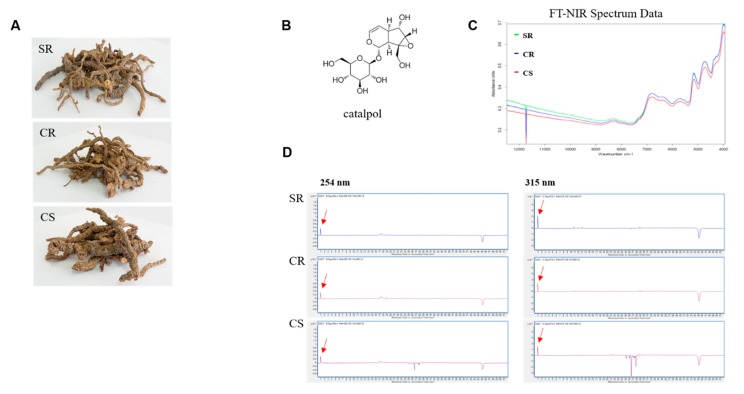
LC/MS analysis of three samples of R. *glutinosa*. (**A**) Three samples of R. glutinosa. SR, CR, and CS. (**B**) The structure of catalpol. (**C**) FT-NIR Spectrum (Green peak. SR, Blue peak. CR, Red peak. CS). (**D**) UV chromatogram of LC/MS for three samples (detection wavelength as 254 and 315 nm), catalpol (red arrow).

**Figure 6 plants-09-00317-f006:**
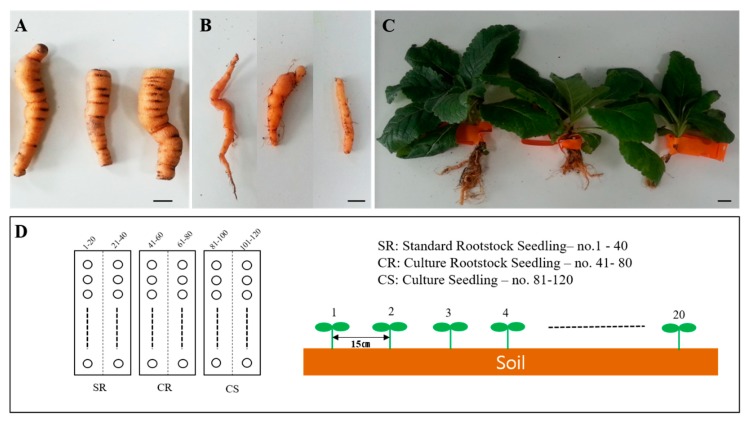
Seed plots for the three types of *R. glutinosa* seedlings and field experiment design. (**A**) SR (standard rootstock seedling). (**B**) CR (culture rootstock seedling). (**C**) CS (culture seedling). Scale bar, 1.0 cm. (**D**) Planting intervals and the number of *R. glutinosa* seedlings planted.

**Table 1 plants-09-00317-t001:** Comparison of survival tendencies and aerial part growth parameters measured for three types of *R.*
*glutinosa* seedlings (SR, CR, and CS).

Growth Characteristics (Aerial Part)	Seedling Types		
	SR	CR	CS
Aliveness	37	36	34
Witheredness	3	4	6
Inflorescence	1.9 ± 0.9	0.0 ± 0.0	0.0 ± 0.0
Peduncle	7.7 ± 3.2	0.0 ± 0.0	0.0 ± 0.0
Base shoot	2.8 ± 1.2 ^a^	3.0 ± 1.3 ^a^	2.4 ± 2.0 ^a^
Rhizome shoot	3.1 ± 2.2 ^b^	4.9 ± 3.5 ^ab^	5.9 ± 3.7 ^a^

a, b: value in same columns with different superscripts are significantly different (*p* < 0.05) with one-way ANOVA and Duncan’s test (alpha = 0.05) using SPSS statistics (IBMCorporation, Ver. 20.0 Armonk, NY, USA).

**Table 2 plants-09-00317-t002:** Comparison of root yields of the different seedling types of *R. glutinosa* (SR, CR, and CS).

Growth Characteristics		Seedling Types		
		SR	CR	CS
Whole part of *R. glutinosa* plant (g)	Total fresh weight	118.0 ± 83.7	294.8±123.0	256.1 ± 133.4
	Total dry weight	22.9 ± 16.8	65.3 ± 26.7	51.9 ± 30.1
Roots (cm^2^) (Herbal medicinal part)	SA	30.3 ± 14.9	46.0 ± 18.7	39.6 ± 18.6
	PA	9.7 ± 4.7	14.6 ± 5.9	13.3 ± 10.9
	Vol	0.4 ± 0.2	0.6 ± 0.3	0.6 ± 0.3

**Table 3 plants-09-00317-t003:** Linearity of the standard curve and detection/quantification limits for the catalpol standard and quantitative analysis of catalpol.

Compound	Catalpol	Samples	Contents of Catalpol (μg/mg) ^d^
R^2 a^	0.992	SR	16.88 (1.688%)
LOD (μg/mg) ^b^	0.031	CR	13.21 (1.321%)
LOQ (μg/mg) ^c^	0.102	CS	8.83 (0.883%)

^a^*y*, peak area; *x*, concentration of the standard (μg/mL); ^a^ R^2^, correlation coefficient for 5 data points in the calibration curves (*n* = 3); ^b^ LOD, limit of detection (S/N = 3); ^c^ LOQ, limit of quantification (S/N = 10); ^d^ Mean ± SD (*n* = 3).

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
