# Peer review of "Verification of the Field Productivity of Rehmannia glutinosa (Gaertn.) DC. Developed Through Optimized In Vitro Culture Method"

_plants, 2020, doi:10.3390/plants9030317_

Round 1

Reviewer 1 Report

General information

Lately, many efforts have been done in order to optimized in vitro culture method. As the authors have noted that the current knowledge focusing on R. glutinosa tissue culturing callus-induction and subsequent regeneration and on seed germination aspects. The achieved results are interesting not only for scientific reasons but also practical. They are an important attempt to show the mass production of the belowground parts of the  CR and CS seedlings, which are used for medicinal purposes, was twice as high as that of the aboveground part of SR. I suggest that the Editors should publish this research, however, prior some issues should be explained.

The manuscript, although prepared well, is not free from mistakes that sometimes lead to confusion. Authors should pay special attention to used plant naming. In addition, the abstract requires reorganization to be more encouraging the reader to read the whole paper. Moreover, much work should be done to improve M&M and Introduction sections, because the current version of them are poor and contain a lot of mistakes.

L3 add space

L23

R. to Rehmannia

Don't start the sentence with shortcuts. This note applies to the entire text!

I also advise you to use linguistic help.

L38

The conclusions are vague and require a closer connection with the objectives.

The Introduction is short and readable. Why did you write so little about catalpol in the introduction? This is a very important substance from the point of view of this plant. This should be emphasized here at work. It contains many interesting and important information. However, some parts requires a reorganization, and its details are listed below.

L42-61

This paragraph is too long should be shortened. He describes matters not related to this research!

L44

R.

L48-49

Delete: R. glutinosa x3

L70-75

“Our”?? Why don't you quote your scientific publication? There are no authors from you in the composition of authors from work 16.

L80-81

This is not precise and not very scientific. Tests should be repeatable, and in this case they are not.

L87

Aeral? Subtteranean part? I think that better will be to use “above and beloveground part” in whle paper

L88-89

This should be in M&M

L108

Enlarge the font in the figure

L112-114

Enlarge the font in the table and whole text under it. +/- SE or SD is provided? Results of the test are correct in SR?

L133

Information about WINRHIZO is not result. Why you

Maybe move M&M chapter as number 2?

L138, 141, 163, 179

Enlarge the font

Author Response

Minor revision

Thank you for your help. Based on your comments, we revised them as much as we can. Please see the revised manuscript which we remarked highlights as Yellow in the texts.

L3 add space

Response: Thank you for this observation and suggestion. We have corrected this accordingly.

L23

R. to Rehmannia Don't start the sentence with shortcuts. This note applies to the entire text! I also advise you to use linguistic help.

Response: We already modified by professional English editors.

L38

The conclusions are vague and require a closer connection with the objectives.

Response: Thank you for the keen observation, conclusion was modified accordingly.

The Introduction is short and readable. Why did you write so little about catalpol in the introduction? This is a very important substance from the point of view of this plant. This should be emphasized here at work. It contains many interesting and important information. However, some parts requires a reorganization, and its details are listed below.

Response: More detail was added about catalpol as advised.

L42-61

This paragraph is too long should be shortened. He describes matters not related to this research!

Response: Contents were reorganized and paragraph revised and split.

L44

R.

Response: Thank you for this observation and suggestion. We have corrected this accordingly.

L48-49

Delete: R. glutinosa x3

Response: Thank you for this observation and suggestion. We have corrected this accordingly.

The correction made does not give meaning to the sentence

L70-75

“Our”?? Why don't you quote your scientific publication? There are no authors from you in the composition of authors from work 16.

Response: Thank you for this observation and suggestion. We have corrected this accordingly.

We changed to 17 reference which Dr. Kang et al published previously.

“Kang, Y.M.; Lee, K.Y.; Kim, M.S.; Choi, J.E.; Moon, B.C. Optimization of In vitro Cultures for Production of Seedling and Rootstock of Rehmannia glutinosa(Gaertn.) DC. J. Agriculture & Life Sci. 2016, 50(5), 81-93”

L80-81

This is not precise and not very scientific. Tests should be repeatable, and in this case they are not.

Response: Thank you for the observation. The statement was rephrased making precise and clearer. The cultivation method is repeatable.

L87

Aeral? Subtteranean part? I think that better will be to use “above and beloveground part” in whle paper

Response: Thank you for this observation and suggestion. Since the term is commonly used in horticulture and crop science, it has not been modified.

L88-89

This should be in M&M

Response: This was addressed as suggested. Thank you.

L108

Enlarge the font in the figure

Response: Thank you for this observation and suggestion. We have corrected this accordingly.

L112-114

Enlarge the font in the table and whole text under it. +/- SE or SD is provided? Results of the test are correct in SR?

Response: Thank you for this observation and suggestion. We have corrected this accordingly.

L133

Information about WINRHIZO is not result. Why you

Maybe move M&M chapter as number 2?

Response: Thank you for this observation and suggestion. We have removed “WinRHIZO”.

Such as “ measured using WinRHIZO,”

L138, 141, 163, 179

Enlarge the font

Response: Thank you for this observation and suggestion. We have corrected this accordingly.

If it is not enough to see in the manuscript on this moment, we will revised during pictorial editing process.

Reviewer 2 Report

The manuscript of an article "Verification of the Field Productivity of Rehmannia glutinosa(Gaertn.) DC. Developed through Optimized In vitro Culture Method" completes the requirements for to be published in the journal Plants. I recommend to accept it in its presented form.

Author Response

Reviewer 2 Response

The manuscript of an article "Verification of the Field Productivity of Rehmannia glutinosa(Gaertn.) DC. Developed through Optimized In vitro Culture Method" completes the requirements for to be published in the journal Plants. I recommend to accept it in its presented form.

Response: Thank you for your review and comments. Our research team will continue to conduct optimized plant tissue culture and verification of field productivity studies.

Reviewer 3 Report

The manuscript entitled “Verification of the field productivity of Rehmannia glutinosa(Gaertn.) DC. developed through optimized in vitro culture method”, the authors showed the differences between R. glutinosa standard rootstock seedlings (SR) and R. glutinosa culture rootstock seedlings (CR) and culture seedlings (CS) under field conditions. Authors attempted to compare the growth characteristics of R. glutinosa plants cultivated in the field using the cultivation method based on their previous studies and R. glutinosa existing in the current market and compared the productivity of the medicinal parts of these plants. The results presented in this study provided a verifiable strategy for the mass production of the medicinal plant R. glutinosa in the field. The data in the manuscript conferred the results with well-controlled and statistically significant figures and easy to follow. I have a few concerns about the data showed, which I think should be addressed before the manuscript is accepted for publication:

Comments and Suggestions to the Authors:

1. Page 2, Line 89, 90 ---
R. glutinosa seedlings were investigated in July. …………………the characteristics of flowers was conducted in September.  The SR, CR, and CS R. glutinosa seedlings were grown under field conditions,  e.g., temperature ? humidity ? the growth conditions ? it should be described in Materials and Methods.

2. Page 3, Figure 2 ---

It should be consistent throughout the MS. Comparison of three types of R. glutinosa seedlings, first place should be SR, the second is CR, and third is CS, just like Figures 1, 3, and 4.

3. Page 4, Figure 3 ---
It will be much easier to understand the results if it directly shows SR, CR, and CS on the figure (likes Figure. 1).

4. Page 6, Figure 5 ---

In panels A and D,  a, b, and c should be omitted, directly shows SR, CR, and CS on the figure; it will be made much more reader-friendly.

Author Response

Comments and Suggestions to the Authors:

Thank you for your help. Based on your comments, we revised them as much as we can.

Please see the revised manuscript which we remarked highlights as Green in the texts.

  1. Page 2, Line 89, 90 ---
    R. glutinosa seedlings were investigated in July. …………………the characteristics of flowers was conducted in September.  The SR, CR, and CS R. glutinosa seedlings were grown under field conditions,  e.g., temperature ? humidity ? the growth conditions ? it should be described in Materials and Methods.

Response: Thank you for this observation and suggestion. The plant growth conditions (Field Average Temperature and Humidity) were assumed not to differ much the year of first investigation.

  1. Page 3, Figure 2 ---

It should be consistent throughout the MS. Comparison of three types of R. glutinosa seedlings, first place should be SR, the second is CR, and third is CS, just like Figures 1, 3, and 4.

Response: Thank you for this suggestion. Corrected accordingly. Please see the Figure 2.

  1. Page 4, Figure 3 ---
    It will be much easier to understand the results if it directly shows SR, CR, and CS on the figure (likes Figure. 1).

Response: Thank you for this suggestion. Corrected accordingly. Please see the Figure 3.

  1. Page 6, Figure 5 ---

In panels A and D,  a, b, and c should be omitted, directly shows SR, CR, and CS on the figure; it will be made much more reader-friendly.

Response: Thank you for this suggestion. Corrected accordi
